Corrected: Publisher Correction

# A space–time tradeoff for implementing a function with master equation dynamics

David H. Wolpert [1,2], Artemy Kolchinsky[1] & Jeremy A. Owen[3]

Master equations are commonly used to model the dynamics of physical systems, including systems that implement single-valued functions like a computer's update step. However, many such functions cannot be implemented by any master equation, even approximately, which raises the question of how they can occur in the real world. Here we show how any function over some "visible" states can be implemented with master equation dynamics—if the dynamics exploits additional, "hidden" states at intermediate times. We also show that any master equation implementing a function can be decomposed into a sequence of "hidden" timesteps, demarcated by changes in what state-to-state transitions have nonzero probability. In many real-world situations there is a cost both for more hidden states and for more hidden timesteps. Accordingly, we derive a "space–time" tradeoff between the number of hidden states and the number of hidden timesteps needed to implement any given function.

[1] Santa Fe Institute, 1399 Hyde Park Road, Santa Fe, NM 87501, USA. [2] Arizona State University, Tempe 85281 AZ, USA. [3] Physics of Living Systems Group, Department of Physics, Massachusetts Institute of Technology, 400 Tech Square, Cambridge, MA 02139, USA. Correspondence and requests for materials should be addressed to D.H.W. (email: dhw@santafe.edu)

Many problems in science and engineering involve understanding how a physical system can implement a given map taking its initial, "input" state to its "output" state at some later time. Often such a map is represented by some stochastic matrix $P$. For example, $P$ may be a conditional distribution that governs the evolution of some naturally occurring system between two particular moments, and we wish to understand what underlying physical process could result in that conditional distribution. Alternatively, $P$ might represent some function $f$ that we wish to implement using a physical process, e.g., $f$ could be the update function of the logical state of a digital computer.

In this paper we uncover constraints on the amounts of various resources that are needed by any system that implements a stochastic matrix $P$. Throughout, we suppose that the underlying dynamics of the system are continuous-time and Markovian. (Such systems are sometimes said to evolve according to a "master equation".) This basic assumption underlies many analyses in stochastic thermodynamics[1–6], and applies to many classical physical systems at the mesoscale, as well as semiclassical approximations of open quantum system with discrete states[7,8]. Master equations also frequently appear in biology, demography, chemistry, computer science, and various other scientific fields. In addition to assuming master equation dynamics, we focus on the case where $P$ represents some single-valued function $f : \mathcal{X} \to \mathcal{X}$ over a finite space of "visible states" $\mathcal{X}$. For example, this would be the case for any physical system that implements an idealized digital device.

The first resource we consider is the number of "hidden states" that are coupled to the states in $\mathcal{X}$ by the master equation at intermediate times within the time interval [0, 1]. The second resource is the number of successive subintervals of [0, 1] which are demarcated by moments when the set of state-to-state transitions allowed by the master equation discontinuously changes. (We refer to each such subinterval as a "hidden timestep".)

In the real world, often it will be costly to have many hidden states and / or many hidden timesteps. For example, increasing the number of hidden states generally requires adding additional storage capacity to the system, e.g., by using additional degrees of freedom. Similarly, increasing the number of hidden timesteps carries a "control cost", i.e., it increases the complexity of the control protocol that is used to drive the dynamics of the system. Moreover, transitions from one timestep to the next, during which the set of allowed state-to-state transition changes, typically require either the raising or dropping of infinite energy barriers between states in some underlying phase space[9–13]. Such operations typically require some minimal amount of time to be carried out. Accordingly, the minimal number of hidden states and the minimal number of hidden timesteps that are required to implement any given function $f$ can be viewed as fundamental "costs of computation" of a function $f$.

Physics has long been interested in the fundamental costs of performing computation and information processing. The most well-known of such costs is "Landauer's bound"[14–18], which states that the erasure of a physical bit, represented by a function $f : \{0,1\} \mapsto 0$, requires the generation of at least $kT\ln 2$ heat when coupled to a heat bath at temperature $T$, assuming the initial value of the bit is uniformly distributed. Recent studies have extended this bound to give the exact minimal amount of heat needed to implement arbitrary functions $f$. These studies have all focused on implementing the given function $f$ with a physical system whose dynamics can be approximated to arbitrary accuracy with master equations[13,19–27]. The two costs of computation proposed here arise, implicitly, in these previous analyses, since that the physical systems considered there all use hidden states.

However, none of these previous papers considered the minimal number of hidden states needed to implement a given function $f$ using master equations. (Rather they typically focused on issues related to thermodynamic reversibility).

In addition, the processes considered in these papers all unfold through a sequence of distinct "timesteps". In any single one of those timesteps, transitions between some pairs of states are allowed to occur while others are blocked, and the set of allowed transitions changes in going from one timestep to the next. Again, despite their use of such hidden timesteps, none of these previous papers considered the minimal number of hidden timesteps needed to implement a function, given a certain number of available hidden states.

Our main results are exact expressions for the minimal number of hidden states needed to implement a single-valued function $f$, and the msinimal number of hidden timesteps needed to implement $f$ given a certain number of hidden states. These results specify a tradeoff between the minimal number of hidden states and the minimal number of hidden timesteps required to implement a given $f$, which is analogous to the "space–time" tradeoffs that arise in the study of various models of computation in computer science. However, here the tradeoff arises from the fundamental mathematical properties of continuous-time Markov processes. Moreover, real-world computers are constructed out of circuits, which are networks of computational elements called gates, each of which carries out a simple function. For circuits, the tradeoff between hidden states and hidden timesteps that we uncover would apply in a "local" sense to the function carried out at each individual gate, whereas computer science has traditionally focused on "global" tradeoffs, concerning the set of all of those functions and of the network coupling them (e.g., the number of required gates or the "depth" of the circuit to compute some complicated $f$).

## Results

**Markov chains and the embedding problem**. We consider finite-state systems evolving under time-inhomogeneous continuous time Markov chains, which in physics are sometimes called "master equations". Such models of the dynamics of systems are fundamental to many fields, e.g., they are very commonly used in stochastic thermodynamics[1,28]. We begin in this subsection by introducing some foundational concepts, which do not involve hidden states or hidden timesteps.

We use calligraphic upper-case letters, such as $\mathcal{X}$ and $\mathcal{Y}$, to indicate state spaces. We focus on systems with a finite state space. We use the term continuous-time Markov chain (CTMC) $T(t, t')$ to refer to a set of transition matrices indexed by $t \leq t'$ which obey the Chapman-Kolmogorov equation $T(t, t') = T(t'', t')T(t, t'')$ for $t'' \in [t, t']$. We use CTMC with finite rates to refer to a CTMC such that the derivatives $\frac{d}{dt} T_{ij}(t, t')$ are well-defined and finite for all states $i$, $j$ and times $t \leq t'$[29]. For a given CTMC $T(t, t')$, we use $T_{ij}(t, t')$ to indicate the particular transition probability from state $j$ at time $t$ to state $i$ at time $t'$. Note that we do not assume time-homogeneous CTMCs, meaning that in general $T(t, t + \tau) \neq T(t', t' + \tau)$. Finally, note that the units of time are arbitrary in our framework, and for convenience we assume that $t=0$ at the beginning of the process and $t=1$ at the end of the process.

The following definition is standard:

**Definition 1.** A stochastic matrix $P$ is called embeddable if $P = T(0, 1)$ for some CTMC $T$ with finite rates.

As it turns out, many stochastic matrices cannot be implemented by any master equation. (The general problem of finding a master equation that implements some given stochastic matrix $P$ is known as the *embedding problem* in the mathematics literature[30–32]). One necessary (but not sufficient) condition

for a stochastic matrix $P$ to be implementable with a master equation is[30],[31],[33]

$$\prod_i P_{ii} \geq \det P > 0. \qquad (1)$$

When $P$ represents a single-valued function $f$ which is not the identity, $\prod_i P_{ii} = 0$, and the conditions of Eq. (1) are not satisfied. Therefore, no non-trivial function can be exactly implemented with a master equation. However, as we show constructively in Supplementary Note 2, all non-invertible functions (e.g., bit erasure, which corresponds to $P = \begin{pmatrix} 1 & 1 \\ 0 & 0 \end{pmatrix}$ can be approximated arbitrarily closely using master equation dynamics. Intuitively, since the determinant of such functions equals 0, they can satisfy Eq. (1) arbitrarily closely.

To account for such cases, we introduce the following definition:

**Definition 2.** A stochastic matrix $P$ is limit-embeddable if there is a sequence of CTMCs with finite rates, $\{T^{(n)}(t, t') : n=1, 2, \ldots\}$, such that

$$P = \lim_{n \to \infty} T^{(n)}(0, 1). \qquad (2)$$

Note that while each $T^{(n)}$ has finite rates, in the limit these rates may go to infinity (this is sometimes called the "quasistatic limit" in physics). This is precisely what happens in the example of (perfect) bit erasure, as shown explicitly in Supplementary Note 1.

We use the term master equation to broadly refer to a CTMC with finite rates, or the limit of a sequence of such CTMCs.

**Definition of space and time costs**. When $P$ represents a (non-identity) invertible function, $\prod_i P_{ii} = 0$, while $\det P$ equals either 1 or $-1$. So the conditions of Eq. (1) are not even infinitesimally close to being satisfied. This means that any (non-identity) invertible function cannot be implemented, even approximately, with a master equation. As an example, the simple bit flip (which corresponds to the stochastic matrix $P = \begin{pmatrix} 0 & 1 \\ 1 & 0 \end{pmatrix}$), cannot be approximated by running any master equation over a two-state system.

How is it possible then that invertible functions can be accurately implemented by actual physical systems that evolve according to a master equation? In this paper, we answer this question by showing that any function $f : \mathcal{X} \to \mathcal{X}$ over a set of visible states $\mathcal{X}$ can be implemented with a master equation—as long as the master equation operates over a sufficiently large state space $\mathcal{Y} \supseteq \mathcal{X}$ that may include additional hidden states, $\mathcal{Y} \backslash \mathcal{X}$. The key idea is that if $\mathcal{Y}$ is large enough, then we can design the dynamics over the entire state $\mathcal{Y}$ to be non-invertible, allowing the determinant condition of Eq. (1) to be obeyed, while at the same time the desired function $f$ is implemented over the subspace $\mathcal{X}$. As an illustration, below we explicitly show below how to implement a bit flip using a master equation over a 3-state system, i.e., a system with one additional hidden state.

The following two definitions formalize what it means for one stochastic matrix to implement another stochastic matrix over a subset of its states. The first is standard.

**Definition 3.** The restriction of a $|\mathcal{Y}| \times |\mathcal{Y}|$ matrix $A$ to the set $\mathcal{X} \subseteq \mathcal{Y}$, indicated as $A_{[\mathcal{X}]}$, is the $|\mathcal{X}| \times |\mathcal{X}|$ submatrix of $A$ formed by only keeping the rows and columns of $A$ corresponding to the elements in $\mathcal{X}$.

In all definitions below, we assume that $P$ is a $|\mathcal{X}| \times |\mathcal{X}|$ stochastic matrix.

**Definition 4.** $M$ implements $P$ with $k$ hidden states if $M$ is a $(|\mathcal{X}| + k) \times (|\mathcal{X}| + k)$ stochastic matrix and $M_{[\mathcal{X}]} = P$.

To see the motivation of Definition 4, imagine that $M$ is a stochastic matrix implemented by some process, and $M_{[\mathcal{X}]} = P$. If at $t = 0$ the process is started in some state $i \in \mathcal{X}$, then the state distribution at the end of the process will be exactly the same as if we ran $P$, i.e., $M_{ji} = P_{ji}$ for all $j \in \mathcal{X}$. Furthermore, because $\sum_{j \in \mathcal{X}} P_{ji} = 1$, $M_{ji} = 0$ for any $i \in X$ and $j \notin \mathcal{X}$ (i.e., for any $j$ which is a hidden state). This means that if the process is started in some $i \in \mathcal{X}$, no probability can "leak" out into the hidden states by the end of the process, although it may pass through them at intermediate times.

The "(hidden) space cost" of $P$ is the minimal number of hidden states required to implement $P$:

**Definition 5.** The (hidden) space cost of $P$, written as $C_{\text{space}}(P)$, is the smallest $k$ such that there exists a limit-embeddable matrix $M$ that implements $P$ with $k$ hidden states.

Consider a CTMC $T$ governing the evolution of a system. As $t$ increases, the set of transitions allowed by the CTMC (that is, the set of states which have $T_{ij}(0, t) > 0$) changes. We wish to identify the number of such changes between $t = 0$ and $t = 1$ as the number of "timesteps" in $T$. To formalize this, we first define the set of "one-step" matrices, which can be implemented by a CTMC which does not undergo any changes in the set of allowed transitions:

**Definition 6.** $P$ is called one-step if $P$ is limit-embeddable with a sequence of CTMCs $\{T^{(n)} : n = 1, 2, \ldots\}$ such that:

1. $T(t, t') := \lim_{n \to \infty} T^{(n)}(t, t')$ exists for all $t, t' \in [0, 1]$;
2. $T(0, t)$ is continuous in $t \in (0, 1]$ and $T(t', 1)$ is continuous in $t' \in [0, 1)$;
3. For all $i$, $j$, either $T_{ij}(0, t) > 0$ for all $t \in (0, 1)$, or $T_{ij}(0, t) = 0$ for all $t \in (0, 1)$.

We note two things about our definition of one-step matrices. First, the precise semi-open interval used in the continuity condition (condition 2) allows discontinuities in $T$ (and therefore in the set of allowed transitions) at the borders of the time interval. Second, we note that the limiting transition matrix $T$ in the above definition is still a CTMC. This is because: (1) a limit of a sequence of stochastic matrices is itself a stochastic matrix, so by definition $T(t, t')$ is a stochastic matrix for all $t$, $t' \in [0, 1]$, and (2) the Chapman-Kolmogorov equation $T(t, t') = T(t'', t')T(t, t'')$ holds (since it holds for each $T^{(n)}$). A canonical example of a one-step map is bit erasure, as demonstrated in Supplementary Note 1. The definition of one-step matrices allows us to formalize the minimal number of timesteps it takes to implement any given $P$:

**Definition 7.** The (hidden) time cost with $k$ hidden states of $P$, written as $C_{\text{time}}(P, k)$, is the minimal number of one-step matrices of dimension $(|\mathcal{X}| + k) \times (|\mathcal{X}| + k)$ whose product implements $P$ with $k$ hidden states.

Note that a product of one-step matrices can be implemented with a CTMC that successively carries out the CTMCs corresponding to each one-step matrix, one after the other. So any stochastic matrix $P$ with finite time cost can be implemented as a single CTMC. Moreover, we can rescale units of time so that that product of one-step matrices is implemented in the unit interval, $t \in [0, 1]$. Note as well that since one-step matrices can have discontinuities at their borders, the adjacency matrix of such a product of one-step matrices can change from one such matrix to the next.

**The space–time tradeoff**. For the rest of this paper, we assume that our stochastic matrix of interest $P$ is 0/1-valued, meaning that it represents a (single-valued) function $f : \mathcal{X} \to \mathcal{X}$. Below, in a slight abuse of previous notation, we will use $C_{\text{space}}(f)$ and $C_{\text{time}}(f, k)$ to refer to the space and time cost of

implementing $f$. Except where otherwise indicated, all proofs are in the Methods section.

As we will show, there is a fundamental tradeoff between the number of available hidden states and the minimal number of timesteps. It will be convenient to present it using some standard terminology[34]. For any function $f : X \to X$, we write fix($f$) for the number of fixed points of $f$, and $|\mathrm{img}(f)|$ for the size of the image of $f$. We also write cycl($f$) for the number of cyclic orbits of $f$, i.e., the number of distinct subsets of $X$ of the form $\{x, f(x), f(f(x)),\dots, x\}$ where $x$ is not a fixed point of $f$ and each element in the subset has a unique inverse under $f$.

We can now state our main result:

**Theorem 1.** For any single-valued function $f$ and number of hidden states $k$,

$$C_{\mathrm{time}}(f,k) = \left\lceil \frac{k + |\mathcal{X}| + \max[\mathrm{cycl}(f) - k, 0] - \mathrm{fix}(f)}{k + |\mathcal{X}| - |\mathrm{img}(f)|} \right\rceil + b(f,k)$$

(3)

where $\lceil \cdot \rceil$ is the ceiling function and $b(f, k)$ equals either zero or one (the precise value of $b(f, k)$ is unknown for some functions).

Several corollaries from this result follow immediately:

**Corollary 2.** For any single-valued function $f$ and number of hidden states $k$,

$$C_{\mathrm{time}}(f,k) \approx \frac{|\mathcal{X}| + \mathrm{cycl}(f) - \mathrm{fix}(f)}{k + |\mathcal{X}| - |\mathrm{img}(f)|} + 1$$

(4)

and

$$C_{\mathrm{time}}(f,k) \leq \frac{1.5 \times |\mathcal{X}|}{k} + 3.$$

(5)

In addition, a "converse" of our main result gives $k_{\min}(f, \tau)$, the minimal number of hidden states $k$ needed to implement $f$, assuming we are allowed to use at most $\tau$ timesteps. The exact equation for $k_{\min}(f, \tau)$ is presented in the Methods section. A simple approximation of that exact converse follows from Corollary 2:

$$k_{\min}(f,\tau) \approx \frac{\mathrm{cycl}(f) + |\mathcal{X}|(2 - \tau) - \mathrm{fix}(f)}{\tau - 1} + |\mathrm{img}(f)|.$$

(6)

Although formulated in terms of time cost, our results have some implications for space cost:

**Corollary 3.** For any non-invertible function $f$, $C_{\mathrm{space}}(f) = 0$. For any invertible $f$ (except the identity), $C_{\mathrm{space}}(f) = 1$.

**Proof.** If $f$ is non-invertible $f$, $|\mathcal{X}| - |\mathrm{img}(f)| \neq 0$, so $C_{\mathrm{time}}(f, 0)$ is finite. Therefore, by definition of $C_{\mathrm{space}}$ and $C_{\mathrm{time}}$, $C_{\mathrm{space}}(f) = 0$. For invertible $f$, the denominator of Eq. (3) is zero if $k = 0$. So while it is possible to implement any such $f$ (except the identity) in a finite number of timesteps if we can use at least one hidden state, it is impossible if we do not have any hidden states, i.e., $C_{\mathrm{space}}(f) = 1$.

Figure 1 illustrates the tradeoff between space cost and time cost for three different functions over $\mathcal{X} = \{0, , 2^{32} - 1\}$. The first function (in blue) is an invertible "cycle" over the state space, computed as $x \mapsto x + 1 \mod 2^{32}$. The second function (in green) is an invertible bitwise NOT operation, in which each element of $\mathcal{X}$ is treated as a 32-bit string and the value of each bit is negated. The third function (in red) is an addition followed by clipping to the maximum value, computed as $x \mapsto \min(x + 2^{16}, 2^{32} - 1)$. Exact results (solid lines), as well as the approximation of Eq. (4) from Corollary 2 (crosses), are shown. These results show that achieving the minimal space costs given in Corollary 3 may result in a very large time cost.

There are two important special cases of our result, which are analyzed in more detail in the Methods section. First, when at least $|\mathrm{img}(f)|$ hidden states are available, any $f$ can be

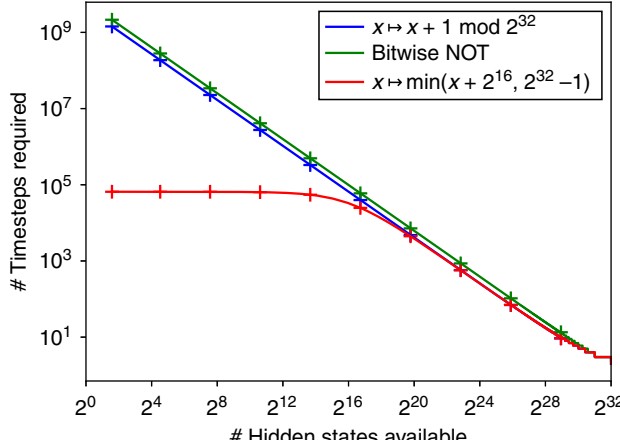

**Fig. 1** The space/time tradeoff for three functions. The domain of all three functions is $\mathcal{X} = \{0, \dots, 2^{32} - 1\}$. Solid lines show exact results, crosses indicate the approximation given by Eq. (4)

implemented in exactly two timesteps. Second, when $f$ is a cyclic permutation and there is one hidden state available, the time cost is exactly $|\mathcal{X}| + 1$.

We emphasize that the proofs of these results (presented in the Methods section) are constructive; for any choice of function $f$ and number of hidden states $k$, this construction gives a sequence of CTMCs with finite rates whose limit implements $f$ while using $k$ hidden states and the minimal number of hidden timesteps for that number of hidden states. These constructions involve explicitly time-inhomogeneous master equations. Indeed, for any time-homogeneous master equation, the set of allowed state transitions can never change, i.e., the only functions $f$ that can be implemented with such a master equation are those that can be implemented in a single timestep. Therefore our demonstrations of functions $f$ with time cost of 2 proves that there are maps that cannot be implemented unless one uses a time-inhomogeneous master equation, no matter how many hidden states are available.

**Explicit constructions saturating the tradeoff**. We now illustrate our results using two examples. These examples use the fact that any idempotent function is one-step, as proved in Theorem 4 in the Methods section. (We remind the reader that a function $f$ is called idempotent if $f(x) = f(f(x))$ for all $x$.)

**Example 1.** Suppose we wish to implement the bit flip function $f : x \mapsto \neg x$ over $\mathcal{X} = \{0, 1\}$. By Corollary 3, since this map is invertible, we need exactly one hidden state to implement it.

We introduce a space of three states $\mathcal{Y} = \{0, 1, 2\}$, and seek a sequence of idempotent functions over $\mathcal{Y}$ that collectively interchange $0 \leftrightarrow 1$. It is straightforward to confirm that our goal is met by the following sequence of idempotent functions:

1.  $\{1, 2\} \mapsto 2, \quad 0 \mapsto 0;$
2.  $\{0, 1\} \mapsto 1, \quad 2 \mapsto 2;$
3.  $\{0, 2\} \mapsto 0, \quad 1 \mapsto 1;$

Each idempotent can be implemented with the one-step CTMC described in Supplementary Note 2. This explicitly shows how to implement a bit flip using one hidden state and three hidden timesteps.

Evaluating Eq. (3) with $k = 1$, $|\mathcal{X}| = |\mathrm{img}(f)| = 2$, cycl($f$) = 1, and fix($f$) = 0 gives

$$C_{\mathrm{time}}(f,1) = 3 + b(f,1).$$

(7)

Thus, the above construction has optimal time cost (and, in this case, $b(f, 1) = 0$).

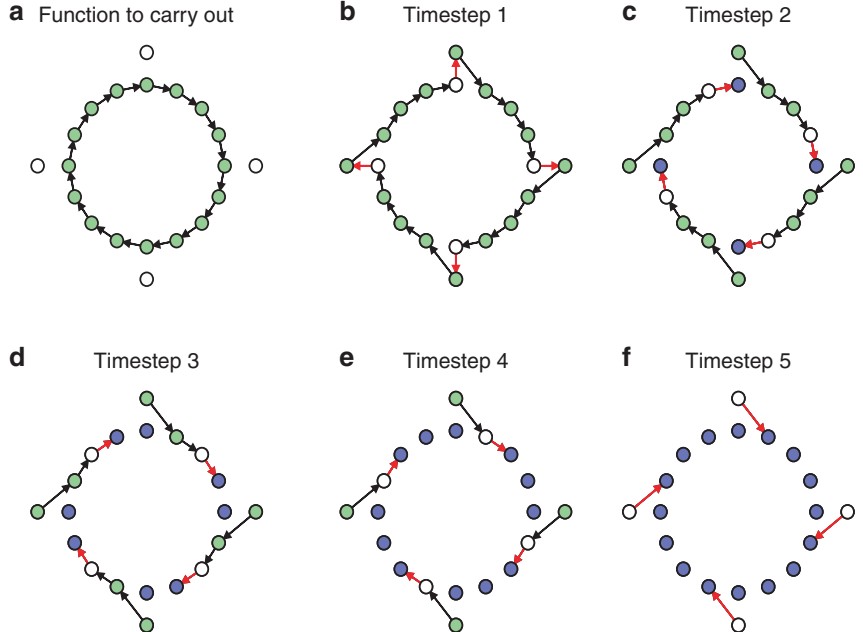

**Fig. 2** Minimal-timestep implementation of a cyclic permutation with 4 hidden states. The implementation carries out the function $f : x \mapsto x + 1 \bmod 16$ over 16 states (green circles in (**a**), using 4 hidden states (white circles in **a**). In all subplots, white nodes indicate states that cannot have any probability, light green nodes with outgoing black arrows indicate states that may have positive probability but are not yet mapped to their final output, and purple nodes indicate states that may have positive probability and have been mapped to their final outputs. Subplots **b**–**f** show the state of the system after each of the 5 timesteps required to carry out $f$, where red arrows indicate the idempotent function carried out in each timestep (in each timestep, any state without outgoing red arrows is mapped to itself)

The following example demonstrates the implementation of a more complicated function, involving a greater number of hidden states.

**Example 2.** Suppose we wish to implement the function

$$f(x) := x + 1 \bmod 16 \tag{8}$$

over $\mathcal{X} = \{1, \ldots, 16\}$. For example, this kind of "cyclic" function may be used to keep track of a clock in a digital computer. Suppose also that 4 hidden states are available, so $\mathcal{Y} = \{1, \ldots, 20\}$. The overall function to carry out, along with the hidden states, are shown in Fig. 2a, along with a sequence of 5 idempotent functions over $\mathcal{Y}$ that carries out $f(x) = x + 1 \bmod 16$ over $\mathcal{X}$. (See caption for details.)

Evaluating Eq. (3) for $k = 4$, $|\mathcal{X}| = |\mathrm{img}(f)| = 16$, $\mathrm{cycl}(f) = 1$, and $\mathrm{fix}(f) = 0$ gives

$$C_{\mathrm{time}}(f, 4) = 5 + b(f, 4). \tag{9}$$

Thus, the above construction of 5 idempotents achieves the minimal time cost for 4 hidden states, and $b(f, 4) = 0$.

See ref. [35] for details on how to decompose more complicated functions into products of idempotent functions.

**Visible states that are coarse-grained macrostates**. Our analysis above concerns scenarios where the full set of states is the union of the set of visible states with the (disjoint) set of hidden states. However, in many real-world physical computers, $f$ is carried out over a set of macrostates that coarse-grain an underlying set of microstates. We call such macrostates "logical states" (logical states are sometimes called the states of the "information bearing degrees of freedom"[36]). The map over the logical states, as specified by $f$, is induced by a master equation evolving over the underlying set of microstates. In such scenarios, we cannot express the full state space as the disjoint union of the logical states with some other "hidden" states, since the logical states are

macrostates. This means that such scenarios cannot be immediately analyzed with our framework.

However, as shown in the Methods, we can generalize our framework to include such maps carried out over logical macrostates, in such a way that scenarios involving disjoint unions of visible and hidden states are just a special case. It turns out that the results of the previous sections apply without any modification, so long as we identify "the number of hidden states" in those results with the difference between the number of microstates and the number of macrostates.

**Example 3.** Suppose we have two quantum dots, each with two possible states, written as $u$ and $w$, respectively, that evolve jointly according to a CTMC[22]. In this scenario, the set of microstates is the set of all four pairs $(u, w)$.

Suppose further that we identify a logical bit with the value of $u$. Then a CTMC over $(u, w)$ will flip the value of the visible state in two (hidden) timesteps if it implements the following sequence of two idempotent functions:

1. $\{(0, 0), (0, 1)\} \mapsto (0, 0); \{(1, 0), (1, 1)\} \mapsto (1, 1)$
2. $\{(0, 0), (1, 0)\} \mapsto (1, 0); \{(1, 1), (0, 1)\} \mapsto (0, 1)$

Since there are four microstates and two logical states (given by the value $u$), this means there are two "hidden states". Thus, applying Theorem 1, with the appropriate change to how $k$ is defined, we conclude that no master equation can implement the bit flip using less than two timesteps. This minimal time cost is in fact achieved by the construction above.

## Discussion

Many single-valued functions from initial to final states cannot be realized by master equation dynamics, even using time-inhomogeneous master equations. In this paper, we show that any single-valued function $f$ over a set of "visible" states $\mathcal{X}$ can be implemented, to arbitrary accuracy—if additional "hidden" states not in $\mathcal{X}$ are coupled to $\mathcal{X}$ by the underlying master equation. We

refer to the minimal number of hidden states needed to implement $f$ as the "space cost" of implementing $f$. In addition, we show that given any function $f$ and number of available hidden states $k$, there is an associated minimal number of timesteps that are needed by any master equation to implement $f$, where we define a "timestep" as a time interval in which the set of allowed transitions between states does not change. We refer to this minimal number of timesteps as the time(step) cost of $f$ for $k$ hidden states.

In this paper, we derive a simple expression for the tradeoff between the space cost and the time cost of any function $f$, a tradeoff which depends on certain algebraic properties of $f$.

We also analyze a generalization of our framework which encompasses scenarios in which visible states are taken to be coarse-grained "logical" macrostates which carry out the desired input-output map, while the hidden states are a subset of the microstates over which the actual master equation unfolds. We show that all of our results regarding space and time costs still apply in this more general setting.

Interestingly, in standard treatments of the thermodynamics of computation, invertible functions can be carried out for free (i.e., while generating no heat), whereas many-to-one maps are viewed as costly. Moreover, noisy (i.e., non-single-valued) stochastic matrices can have lower thermodynamic cost than invertible single-valued ones, in the sense that the minimal free energy required to implement them can actually be negative[13,26,27]. In contrast, when considering the number of hidden states required to implement a computation, it is many-to-one maps that are free, and single-valued invertible ones that are costly. Furthermore, as shown in our companion paper[37], noisy maps may require more hidden states to implement than single-valued ones. Thus, the relative benefits of many-to-one, invertible, and noisy maps are exactly opposite when considering thermodynamic costs versus space and time costs.

The results derived in this paper are independent of considerations like whether detailed balance holds, how many thermal reservoirs the system is connected to, the amount of entropy production incurred by the stochastic process, etc. Nonetheless, in Supplementary Note 2, we show by construction that one can implement any $f$ using the minimal number of hidden states and timesteps using a master equation that (1) obeys detailed balance, (2) evolves probabilities in a continuous manner, and (3) has vanishing entropy production, i.e., is thermodynamically reversible. The latter two properties are satisfied when the equilibrium distribution of the master equation at $t = 0$ (determined by the choice of $q$ in the construction in Supplementary Note 2) coincides with the initial distribution over states (this and related issues are studied further in ref. [38]).

This demonstrates that the implementation costs we consider are novel, and independent from energetic costs like heat and work that are traditionally studied in thermodynamics of computation. Indeed, while our analysis is driven by physical motivations, it applies broadly to any field in which master equation dynamics play an important role.

Our analysis suggests several important directions for future work:

1. Here, we focused on tradeoffs involved in implementing single-valued functions, but typical real-world digital devices cannot achieve perfect accuracy—they will always have some noise. An important line for future work is to extend our analysis to the space and timesteps tradeoffs for the case of arbitrary $P$, including non-single-valued maps. Some preliminary work related to this issue is presented in ref. [37], where we present bounds (not exact equalities) on the space cost of arbitrary stochastic matrices. As discussed there, those space cost bounds have some implications for bounds (again, inexact) on the time cost.

An associated goal is to analyze the tradeoffs for implementing a given $f$ up to some accuracy $\epsilon$. In this setting, a quantity of fundamental interest may be the maximal size $E_{max}$ of allowed energy barriers, which will determine how small entries of the rate matrix can be made. In particular, it is of interest to investigate the coupled tradeoffs between space cost, time cost (appropriately generalized), $\epsilon$, and $E_{max}$, and show how these reduce to a two-way tradeoff between space cost and time cost in the appropriate limit. (The analysis done here corresponds to the case where $\epsilon = 0$ and $E_{max} = \infty$.)

2. Our results quantify the space/time tradeoff under the "best-case" scenario, where there are no restrictions on the dynamical processes available to an engineer who is constructing a system to carry out some map. In particular, we assume that a system's dynamics can be sufficiently finely controlled so as to produce any desired idempotent function. In real-world situations, however, it is likely that the set of idempotent functions that can be engineered into a system will be a tiny fraction of the total number possible, $\sum_{i=1}^{|\mathcal{X}|} \binom{|\mathcal{X}|}{i} i^{|\mathcal{X}|-i}$[39]. (This already exceeds a trillion if there are just 4 bits, so that $|\mathcal{X}| = 16$.) We perform an initial exploration of the consequences of such restrictions in Supplementary Note 6, but there is significant scope for future study of related tradeoffs.

3. Future work will also involve extending our framework to evaluate space and timestep tradeoffs for functions over infinite state spaces, in particular, to extend our results to Turing machines. (See[37] for preliminary analysis of the space costs of implementing noisy stochastic matrices over countably infinite spaces).

## Methods

Our proofs are fully constructive. At a high level, the construction can be summarized as follows:

(1) Adapting an existing result in semigroup theory[35], we find the minimal (length) sequence of idempotent functions on a state space $\mathcal{Y}$ ($|\mathcal{Y}| = |\mathcal{X}| + k$) whose composition equals $f$ when restricted to $\mathcal{X} \subseteq \mathcal{Y}$.

(2) We show that any idempotent function is one-step, by explicitly specifying (see Supplementary Note 2) rate matrices and a limiting procedure for limit-embedding any idempotent function. Thus, the length of the minimal sequence of idempotent functions whose composition implements $f$ with $k$ hidden states, as found in step (1), is an upper bound on $C_{time}(f, k)$.

(3) We show that if a CTMC implements $f$ with $k$ hidden states and $\ell$ timesteps, then there must exist $\ell$ idempotent functions whose composition implements $f$ with $k$ hidden states. Together with step (1) and (2), this means that $C_{time}(f, k)$ is exactly equal to the minimal number of idempotents whose composition implements $f$ with $k$ hidden states.

(4) Therefore, by chaining together the CTMCs implementing the idempotent functions in the decomposition we found in step (1), we construct a CTMC that implements $f$ while achieving our space and timestep bounds.

The rest of this section presents the details.

**Time cost and idempotent functions**. Although our definitions apply to any stochastic matrix $P$, our results all concern 0/1-valued stochastic matrices representing single-valued functions $f$. This is because there is a special relationship between one-step matrices that represent single-valued functions and idempotent functions, a relationship that in turn allows us to apply a result from semigroup theory to calculate time cost—but only of single-valued functions.

We begin with the following, which is proved in Supplementary Note 2.

**Theorem 4.** Any idempotent function over a finite $\mathcal{X}$ is one-step.

Theorem 4 means that we can get an upper bound on the time cost of a single-valued matrix $P$ over a finite $\mathcal{Y}$ by finding the minimal number of idempotent functions that equals $P$. It turns out that this bound is tight, as proved in Supplementary Note 4:

**Lemma 5.** Suppose the stochastic matrix $P$ over $\mathcal{Y} \supseteq \mathcal{X}$ has time cost $\ell$ and the restriction of $P$ to $\mathcal{X}$ is a function $f : \mathcal{X} \to \mathcal{X}$. Then there is a product of $\ell$ idempotent functions over $\mathcal{X}$ whose restriction to $\mathcal{X}$ equals $f$.

By combining these results, we simplify the calculation of the time cost of a function $f$ to the problem of finding a minimal set of idempotent functions whose product is $f$:

**Corollary 6.** The time cost of any function $f$ with $k$ hidden states is the minimal number of idempotents over $\mathcal{Y} = \mathcal{X} \cup \{1, \ldots, k\}$ such that the product of those idempotents equals $f$ when restricted to $\mathcal{X}$.

Idempotent functions have been extensively studied in semigroup theory[35,40–43]. Corollary 6 allows us to exploit results from those studies to calculate the time cost (to within 1) for any function. In particular, we will use the following Theorem, proved in[35] in an analysis of different issues:

**Theorem 7.** Let $f : \mathcal{X} \to \mathcal{X}$ be non-invertible. Then

$$C_{\text{time}}(f, 0) = \left\lceil \frac{|\mathcal{X}| + \text{cycl}(f) - \text{fix}(f)}{|\mathcal{X}| - |\text{img}(f)|} \right\rceil + b(f, 0). \quad (10)$$

where $b(f, 0)$ equals either zero or one.

The expression for $b(f, 0)$ is not easy to calculate, though some sufficient conditions for $b(f, 0) = 0$ are known[35].

**Proofs of our main results. Theorem 1.** Let $f : \mathcal{X} \to \mathcal{X}$. For any number of hidden states $k > 0$, the time cost is

$$C_{\text{time}}(f, k) = \left\lceil \frac{k + |\mathcal{X}| + \max[\text{cycl}(f) - k, 0] - \text{fix}(f)}{k + |\mathcal{X}| - |\text{img}(f)|} \right\rceil + b(f, k) \quad (11)$$

where $b(f, k)$ equals either zero or one.

**Proof.** Let $\mathcal{Y} = \mathcal{X} \cup \mathcal{Z}$ where $\mathcal{Z} \cap \mathcal{X} = \emptyset$ and $|\mathcal{Z}| = k$. By definition $C_{\text{time}}(f, k)$ is the minimum of $C_{\text{time}}(g, 0)$ over all non-invertible functions $g : \mathcal{Y} \to \mathcal{Y}$ that equal $f$ when restricted to $\mathcal{X}$. Moreover, by Theorem 7,

$$C_{\text{time}}(g, 0) = \left\lceil \frac{|\mathcal{Y}| + \text{cycl}(g) - \text{fix}(g)}{|\mathcal{Y}| - |\text{img}(g)|} \right\rceil + b(g, 0) \quad (12)$$

Due to the constraint that $g(x) = f(x)$ for all $x \in \mathcal{X}$, our problem is to determine the optimal behavior of $g$ over $\mathcal{Z}$. For any fixed $\text{img}(g)$, this means finding the $g$ that minimizes $\text{cycl}(g) - \text{fix}(g)$. Since $\text{img}(f) \subseteq \mathcal{X}$, the constraint tells us that there are no cyclic orbits of $g$ that include both elements of $\mathcal{X}$ and elements of $\mathcal{Z}$. So all cyclic orbits of $g$ either stay wholly within $\mathcal{Z}$ or wholly within $\mathcal{X}$. Moreover changing $g$ so that all elements of a cyclic orbit $\Omega$ lying wholly in $\mathcal{Z}$ become fixed points of $g$ does not violate the constraint and reduces the time cost. Therefore under the optimal $g$, all $z \in \mathcal{Z}$ must either be fixed points or get mapped into $\text{img}(f)$.

Our problem then reduces to determining precisely where $g$ should map those elements in $\mathcal{Z}$ it sends into $\text{img}(f)$. To determine this, note that $g$ might map an element of $\mathcal{Z}$ into an $x$ that lies in a cyclic orbit of $f$, $\Omega$. If that happens, $\Omega$ will not be a cyclic orbit of $g$—and so the time cost will be reduced. Thus, to ensure that $\text{cycl}(g)$ is minimal, we can assume that all elements of $\mathcal{Z}$ that are not fixed points of $g$ get mapped into $\text{img}(f)$, with as many as possible being mapped into cyclic orbits of $f$.

Suppose $g$ sends $m \leq k$ of the hidden states into the image of $f$, where each can be used to "destroy" a cyclic orbit of $f$ (until there are none left, if possible). The remaining $k - m$ hidden states are fixed points of $g$. Moreover, since $g(\mathcal{X}) = \text{img}(f)$,

$$|\text{img}(g)| = |\text{img}(f)| + k - m. \quad (13)$$

So using Theorem 7,

$$C_{\text{time}}(g, 0) = \left\lceil \frac{m + |\mathcal{X}| + \max[\text{cycl}(f) - m, 0] - \text{fix}(f)}{m + |\mathcal{X}| - |\text{img}(f)|} \right\rceil + b(g, 0). \quad (14)$$

The quantity inside the ceiling function is minimized if $m$ is as large as possible, which establishes the result once we take $b(f, k) := b(g, 0)$ for the $g$ which has $m = k$ and smallest $b(g, 0)$.

**Corollary 8.** For any $f$ and number of hidden states $k$,

$$C_{\text{time}}(f, k) \approx \frac{|\mathcal{X}| + \text{cycl}(f) - \text{fix}(f)}{k + |\mathcal{X}| - |\text{img}(f)|} + 1. \quad (15)$$

to within 2 timesteps.

**Proof.** Whenever $k \leq \text{cycl}(f)$, the approximation of Eq. (15) holds up to the accuracy of 1 timestep, since the $+1$ term accounts for error due to both the ceiling function and the term $b(f, k) \in \{0, 1\}$. The equivalent approximation for $k > \text{cycl}(f)$ is

$$\frac{k + |\mathcal{X}| - \text{fix}(f)}{k + |\mathcal{X}| - |\text{img}(f)|} + 1, \quad (16)$$

and also holds up to the accuracy of 1 timestep. However, when $k > \text{cycl}(f)$, Eq. (16) will never be more than 1 greater than Eq. (15). To see why, note that Eq. (15) subtracted from Eq. (16) gives

$$\frac{k - \text{cycl}(f)}{k + |\mathcal{X}| - |\text{img}(f)|}. \quad (17)$$

For $k > \text{cycl}(f)$, this quantity is bigger than 0. At the same time, Eq. (17) is always smaller than 1, since the numerator is smaller than the denominator (observe that $|\mathcal{X}| - |\text{img}(f)| \geq 0$).

**Corollary 9.** For any $f : \mathcal{X} \to \mathcal{X}$,

$$C_{\text{time}}(f, k) \leq \frac{1.5 \times |\mathcal{X}|}{k} + 3. \quad (18)$$

**Proof.** First, assume $|\mathcal{X}|$ is even and consider some function $f^*$ which has $f^*(f^*(x)) = x$ and $f^*(x) \neq x$ for all $x \in \mathcal{X}$. One can verify that for this $f$, $\text{cycl}(f) = |\mathcal{X}|/2$, $\text{fix}(f) = 0$, and $|\text{img}(f)| = |\mathcal{X}|$, and that these values maximize the approximation to the time cost given by Corollary 8. This approximation is accurate to within 2 timesteps, which implies the bound

$$C_{\text{time}}(f, k) \leq \frac{1.5 \times |\mathcal{X}|}{k} + 3. \quad (19)$$

If $|\mathcal{X}|$ is odd, the maximum number of cyclic orbits is $(|\mathcal{X}| - 1)/2$, so the above upper bound can be tightened by $1/(2k)$.

**Corollary 10.** Let $\tau > 3$ and define

$$k^* := \left\lceil \frac{\text{cycl}(f) - |\mathcal{X}|(\tau - 3) - \text{fix}(f)}{(\tau - 2)} \right\rceil + |\text{img}(f)|$$
$$k^{**} := \left\lceil \frac{|\text{img}(f)|(\tau - 2) - \text{fix}(f)}{(\tau - 3)} \right\rceil - |\mathcal{X}|. \quad (20)$$

We can implement $f$ in $\tau$ timesteps if we have at least $k$ hidden states, where

$$k = \begin{cases} \max[k^*, 0] & \text{if } k^* < \text{cycl}(f) \\ \max[k^{**}, 0] & \text{otherwise} \end{cases} \quad (21)$$

**Proof.** Since $b(f, k)$ is 0 or 1, by Theorem 1 we know that we can implement $f$ if $\tau$ and $k$ obey

$$\tau \geq \left\lceil \frac{k + |\mathcal{X}| + \max[\text{cycl}(f) - k, 0] - \text{fix}(f)}{k + |\mathcal{X}| - |\text{img}(f)|} \right\rceil + 1. \quad (22)$$

This inequality will hold if

$$\tau \geq \frac{k + |\mathcal{X}| + \max[\text{cycl}(f) - k, 0] - \text{fix}(f)}{k + |\mathcal{X}| - |\text{img}(f)|} + 2. \quad (23)$$

The RHS is non-increasing in $k$. So we can implement $f$ in $\tau$ timesteps, as desired, if $k$ is the smallest integer that obeys the inequality.

First hypothesize that the smallest such $n$ is less than $\text{cycl}(f)$. In this case $\max[\text{cycl}(f) - k, 0] = \text{cycl}(f) - k$. So our bound becomes

$$\tau \geq \frac{k + |\mathcal{X}| + \text{cycl}(f) - k - \text{fix}(f)}{k + |\mathcal{X}| - |\text{img}(f)|} + 2, \quad (24)$$

which is saturated if

$$k = \frac{|\mathcal{X}|(3 - \tau) - \text{fix}(f) + \text{cycl}(f)}{(\tau - 2)} + |\text{img}(f)|. \quad (25)$$

If instead the least $k$ that obeys our inequality is greater than or equal to $\text{cycl}(f)$, then our bound becomes

$$\tau \geq \frac{k + |\mathcal{X}| - \text{fix}(f)}{k + |\mathcal{X}| - |\text{img}(f)|} + 2, \quad (26)$$

which is saturated if

$$k = \frac{|\text{img}(f)|(\tau - 2) - \text{fix}(f)}{\tau - 3} - |\mathcal{X}|. \quad (27)$$

The fact that $k$ must be a nonnegative integer completes the proof.

**Corollary 11.** Any $f$ can be implemented in two timesteps, as long as $|\text{img}(f)|$ hidden states are available.

**Proof.** Consider an implementation of $f$ when $k = |\text{img}(f)|$ hidden states are available. Index the states in $\mathcal{Y}$ using $1, \ldots, |\mathcal{X}|$ for the states in $\mathcal{X}$ and $|\mathcal{X}| + 1, \ldots, |\mathcal{X}| + k$ for the hidden states. The function $f$ can then be implemented as a product of two idempotents:

1. In the first step, for each $x \in \mathcal{X}$, both $x$ and $k + f(x)$ are mapped to $k + f(x)$;
2. In the second step, for each $x' \in \text{img}(f)$, both $x'$ and $k + x'$ are mapped to $x'$.

**Corollary 12.** If $f : \mathcal{X} \to \mathcal{X}$ is a cyclic permutation with no fixed points and there is one hidden state available, then the time cost is $|\mathcal{X}| + 1$.

**Proof.** Theorem 8 tells us that the time cost of $f$ is $|\mathcal{X}| + 1$ or $|\mathcal{X}| + 2$. To show that it is in fact $|\mathcal{X}| + 1$, write the states of $\mathcal{X}$ as $\{1, 2, \ldots, |\mathcal{X}|\}$, with the single hidden state written as $|\mathcal{X}| + 1$. Assume without loss of generality that the states are numbered so that $f(i) = i + 1 \mod |\mathcal{X}|$. Then have the first idempotent function send $\{|\mathcal{X}|, |\mathcal{X}| + 1\} \mapsto |\mathcal{X}| + 1$ (leaving all other states fixed), the second function send $\{|\mathcal{X}| - 1, |\mathcal{X}|\} \mapsto |\mathcal{X}|$ (leaving all other states fixed), etc., up to the $|\mathcal{X}|$'th idempotent function, which sends $\{1, 2\} \mapsto 2$ (leaving all other states fixed). Then have the last idempotent function send $\{1, |\mathcal{X}| + 1\} \mapsto 1$ (leaving all other states fixed). It is easy to verify that this sequence of $|\mathcal{X}| + 1$ idempotent functions performs the cyclic orbit, as claimed.

It is straightforward to use the proof technique of Corollary 12 to show that, in Theorem 1, $b(f, 1) = 0$ for any invertible $f$.

**Extension to allow visible states to be coarse-grained macrostates**. If the visible states are identified with a set of macrostates given by coarse-graining an underlying set of microstates, then the framework introduced above, where $\mathcal{X} \subseteq \mathcal{Y}$, does not directly apply. It turns out though that we can generalize that framework

to apply to such scenarios as well. To show how we start with the following definition:

**Definition 8.** A function $\hat{f} : \mathcal{Z} \to \mathcal{Z}$ can be implemented with $n$ microstates and $\ell$ timesteps if and only if there exists a set $\mathcal{Y}$ with $n$ states and a partial function $g : \mathcal{Y} \to \mathcal{Z}$ such that

1. $\mathrm{img}(g) = \mathcal{Z}$,
2. there exists a stochastic matrix $M$ over $\mathcal{Y}$ which is a product of $\ell$ one-step matrices,
3. for all $i \in \mathrm{dom}(g)$, $\sum_{j \in g^{-1}(\hat{f}(g(i)))} M_{ji} = 1$.

The minimal number $n$ such that $\hat{f}$ can be implemented with $n$ microstates (for some associated $g$ and $M$, and any number of timesteps) we call the microspace cost of $\hat{f}$.

Note that we allow the coarse-graining function to be partially specified, meaning that some microstates may have an undefined corresponding macrostate. Nonetheless, condition 1 in Definition 8 provides that each macrostate is mapped to by at least one microstate. An example of Definition 8 is given by the class of scenarios analyzed in the previous sections, in which $\mathcal{Z} = \mathcal{X} \subseteq \mathcal{Y}$, $g(x) = x$ for all $x \in \mathcal{X}$ and is undefined otherwise, and the elements $\mathcal{Y} \backslash \mathcal{X}$ are referred to as hidden states. Note, however, that in Definition 8, we specify a number of microstates, rather than a number of hidden states. As illustrated in Example 3, this flexibility allows us to consider scenarios in which each $z \in \mathcal{Z}$ is not a single element of the full space $\mathcal{Y}$, but rather a coarse-grained macrostate of $\mathcal{Y}$.

**Definition 9.** Let $\hat{f}$ be a single-valued function over $\mathcal{Z}$ that can be implemented with $n$ microstates. Then we say that the (hidden) time(step) cost of $\hat{f}$ with $n$ microstates is the minimal number $\ell$ such that $\hat{f}$ can be implemented with $n$ microstates.

The minimization in Definition 9 is implicitly over the set of partial macrostates, the matrix $M$, and the function $g$.

The proof of the following Theorem is left for the Supplementary Information.

**Theorem 13.** Assume $\hat{f} : \mathcal{Z} \to \mathcal{Z}$ can be implemented with $n$ microstates and $\ell$ timesteps. Then there is a stochastic matrix $W$ over a set of $n$ states $\mathcal{Y}$, a subset $\mathcal{X} \subseteq \mathcal{Y}$ with $|\mathcal{X}| = |\mathcal{Z}|$, and a one-to-one mapping $\omega : \mathcal{Z} \to \mathcal{X}$ such that

1. $W$ is a product of $\ell$ one-step matrices
2. The restriction of $W$ to $\mathcal{X}$ carries out the function $f(x) := \omega(\hat{f}(\omega^{-1}(x)))$

We are finally ready to prove the equivalence between time cost as defined in previous sections, and time cost for computations over coarse-grained spaces.

**Corollary 14.** Consider a system with microstate space $\mathcal{Y}$. The hidden time cost of a function $\hat{f}$ over a coarse-grained space $\mathcal{Z}$ with $n$ microstates equals the hidden time cost of $\hat{f}$ (up to a one-to-one mapping between $\mathcal{Z}$ and $\mathcal{X} \subseteq \mathcal{Y}$) with $n - |\mathcal{Z}|$ hidden states.

**Proof.** Let $\ell$ indicate the time cost of $\hat{f} : \mathcal{Z} \to \mathcal{Z}$ with $n$ microstates, and let $M$ be a stochastic matrix that achieves this (microstates-based) time cost. Similarly, let $\ell'$ indicate the time cost of carrying out $\hat{f}$ over $\mathcal{X} \subseteq \mathcal{Y}$ (up to a one-to-one mapping between $\mathcal{Z}$ and $\mathcal{X}$, which we call $\omega : \mathcal{Z} \to \mathcal{X}$) with $n - |\mathcal{Z}|$ hidden states, and let $M'$ be a stochastic matrix that achieves this (hidden-states-based) time cost. We prove that $\ell = \ell'$ by proving the two inequalities, $\ell \leq \ell'$ and $\ell' \leq \ell$.

By Theorem 13, it must be that there exists an implementation of $\hat{f}$ over $\mathcal{X}$ with $n - |\mathcal{Z}|$ hidden states and $\ell$ timesteps. Thus, $\ell' \leq \ell$. We can also show that $\ell \leq \ell'$. To do so, define the coarse-graining function $g(x) := \omega^{-1}(x)$ for all $x \in \mathrm{img}(\omega)$, and $g(x)$ undefined for all $x \notin \mathrm{img}(\omega)$. It is easy to verify that $M$ and $g$ satisfies the conditions of Definition 8 with $n$ microstates and $\ell'$ timesteps. Thus, $\ell \leq \ell'$.

## Data availability
No datasets were generated or analysed during the current study.

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

## Acknowledgements

We would like to thank the Santa Fe Institute for helping to support this research. This paper was made possible through the support of Grant No. FQXi-RFP-1622 from the FQXi foundation, and Grant No. CHE-1648973 from the U.S. National Science Foundation.

## Author contributions

D.H.W. came up with the project; the research was done by A.K., J.A.O., and D.H.W.; the writing was done by A.K., J.A.O., and D.H.W.

## Additional information

**Competing interests:** The authors declare no competing interests.

