## [Peer Review File · Nature Communications]

Reviewers' comments:

Reviewer #1 (Remarks to the Author):

This paper is concerned with the ability to implement deterministic logical operations using inhomogeneous Markovian master equations (IMMEs). This type of problems belongs to the field of Brownian computing and are of relevance in a large variety of contexts in biology, physics, chemistry and engineering. Recent experiments have also been performed in recent years, mostly with colloidal particles or electrons in quantum dots. This field received a lot of attention in recent years also because stochastic thermodynamics, by building a thermodynamics on top of the dynamics, enables to study the energetic cost of computation as well as the various trade-offs involved between energy cost, accuracy and speed of the logical operations.

Many previous studies focused on the erasure of a bit of information. In this paper the authors ask the more general question whether any logical operation can be implemented by IMMEs. They find that this is not the case but propose as a workaround to introduce additional hidden states which coupled to the visible ones during the operation but remain globally unaltered in the input and output. The adjacency matrix of the rate matrix is also allowed to change along the the time-steps implementing the logical operation. The minimal number of hidden states necessary to implement the logical operation is defined as the space cost of the operation. The minimal number of time steps necessary to realize the operation is the timestep cost.

The main result of the paper is an explicit formula for the timesteps cost necessary to implement an operation with a given number of hidden states, and vice versa, for the space cost to implement the operation in a given number of timesteps. Three examples are worked out in detail.

The paper addresses in a novel way the important issue of the trade-offs between time and space in Brownian computing. The results are rigorous, non-trivial and convincing. The paper is also very well written and clear. I also think that this study is of interest to many communities, from physics to computer science and biology. I therefore strongly support the paper for publication in Nature Communication.

Reviewer #2 (Remarks to the Author):

The paper is concerned with the implementation of certain deterministic transformations on a state space by means of time-inhomogeneous master equations. To this aim, the authors suggest considering an enlarged state space. They further introduce a notion of hidden time step, to be understood as time intervals over which the set of allowed transitions do not change. They finally provide a relationship between the number of extra dimensions or hidden states and the minimal number of hidden time steps.

I do not find that the results, their actual consistency, and relevance are properly spelled out and supported. In particular, two main points are unclear

The first point is about connection and relevance to physical systems.

The authors draw their motivation from research work in thermodynamics, to justify considering master equations, pointing to possible relevance in modeling physical systems. Actually, the title reads "The minimal hidden computer needed to implement a visible computation", which does not seem to correctly advertise the content.

The authors further state that logically invertible functions can always be implemented via a master equation, and consider simple examples in this respect. Though it sounds plausible and not

surprising (master equations are typically non-invertible, but one can reach this result by enlarging the state space), a constructive proof is needed to support the result and make it actually useful. If I get it correctly, the result is only indirectly inferred building on the notion of time step.

Moreover, the claim that the considered space/time tradeoff or relationship is of some physical relevance is actually not supported, though mentioned as one of the basic motivations of the paper. The authors suggest that their statements comply with previous work (Refs.[23-25]), e.g. when considering Example 1. But other new physical examples showing predictive power of the approach are missing. Reference is always to quantum dots. The result remains too abstract. Additionally, how to fix the initial conditions on the extra degrees of freedom? Is there any estimate on the extra dimensions needed?

The second point is about the treatment of time-inhomogeneous situations as mentioned.

Despite mentioning time-inhomogeneous master equations, there is no actual connection to this topic. The basic result on which the paper stands is drawn from Ref.[42], which refers to semigroups (actually in a framework quite different from the standard one for time-homogeneous master equations considered for physical systems, so that in any case the point should be clearly spelled out to address a broad readership). The extension to the time-inhomogeneous case is certainly not a trivial step, but it is not mentioned. Indeed time is never mentioned, nor taken explicitly into account. When considering the composition of time-dependent maps, also continuity becomes an important issue in order to comply with the dynamics of a physical system, so that this is not a trivial point. The authors should definitely clarify this point.

On these grounds, I would not recommend the paper on publication on Nature Communications. I find the paper could be suitable for a more specialized journal and should, in any case, better clarify the connection to the master equation formalisms and more specifically the issue of time-inhomogeneity.

Reviewer #3 (Remarks to the Author):

A typical stochastic matrix that maps an initial state at time $t=0$ in a finite state at time $t=1$ cannot be implemented by a simple Markovian dynamics. In this paper the authors shows that if one introduces hidden variables in the configuration space and hidden time steps, then a Markovian dynamics exists and its computational cost is estimated. Simple examples of information erase and permutation illustrate the different possibilities.

I am not an expert of the field and I cannot judge the originality and the impact of the paper. I can say that the authors make their point and the paper is clear to understand.

Reviewer #4 (Remarks to the Author):

In this manuscript Wolpert et al. discuss the "space cost" and "time cost" in order to implement a single valued function with a master equation. While the general subject of the manuscript is interesting, I find the manuscript really hard to follow. The main problem in my view is the lack of clear definitions for many of the quantities introduced by the authors. I would expect that even specialists would have a hard time reading this manuscript. I stopped after the end of Sec. II A. Several definitions remain unclear to me.

There is lot of jargon and almost nothing is defining in an equation.

I think that the manuscript needs a very serious rewriting. It remains to be seen whether their results deserve publication in Nature Communications. Specific considerations follow.

- 1) Overall the manuscript is really heavy on jargon. I would recommend that the authors eliminate some of it. For example, you can eliminate "time-inhomogeneous master equation" and simply say "master equation" in the introduction.
- 2) In the introduction, it is crucial that the authors clearly explain what is a logically invertible function and a logically non-invertible function. Do you need the word "logically"?
- 3) In the introduction, "Given this" and "Motivated by this" might not be the best way to start a paragraph.
- 4) In the introduction, change "out discussion" to "our discussion".
- 5) On page 2, in the paragraph starting with "Both these" the authors say that "changing energy barriers involves doing thermodynamic work". That is not correct, changing the energy of a state does involve work but changing the energy barriers between states does not. Also the statement "everything else being equal" is quite unclear.
- 6) "minimal number of hidden states of hidden time-steps" looks wrong.
- 7) The title of subsection IIIA is not good since the authors also define space cost in this part. Adding structure to the manuscript by dividing this subsection will probably improve presentation.
- 8) All definitions in Sec IIIA would need an equation, otherwise it is really hard to follow: after a page, one is left with a lot of different names that cannot be identified with an equation with a number. Examples of things that have to be defined in equations with numbers are: adjacency matrix, embedable, limit-embedable, space cost, restriction of a matrix, stable matrix, and time-step cost.

Reviewer 2 comments and responses:

The paper is concerned with the implementation of certain deterministic transformations on a state space by means of time-inhomogeneous master equations. To this aim, the authors suggest considering an enlarged state space. They further introduce a notion of hidden time step, to be understood as time intervals over which the set of allowed transitions do not change. They finally provide a relationship between the number of extra dimensions or hidden states and the minimal number of hidden time steps.

I do not find that the results, their actual consistency, and relevance are properly spelled out and supported. In particular, two main points are unclear

The first point is about connection and relevance to physical systems.

The authors draw their motivation from research work in thermodynamics, to justify considering master equations, pointing to possible relevance in modeling physical systems. Actually, the title reads "The minimal hidden computer needed to implement a visible computation", which does not seem to correctly advertise the content.

We appreciate the reviewer's suggestion that the title did not correctly advertise the content. We have changed the title to "A space/time tradeoff for implementing a function with master equation dynamics", which we believe more accurately reflects the main contribution of the paper.

Furthermore, while our results relate to statistical physics, they are more general. We now clarify this point.

The authors further state that logically invertible functions can always be implemented via a master equation, and consider simple examples in this respect. Though it sounds plausible and not surprising (master equations are typically non-invertible, but one can reach this result by enlarging the state space), a constructive proof is needed to support the result and make it actually useful. If I get it correctly, the result is only indirectly inferred building on the notion of time step.

Our approach is in fact fully constructive, but we understand that this was not made sufficiently clear in the original version of the manuscript. To address this issue, we have modified the manuscript in the following ways:

1. Appendix B now provides an explicit construction of a master equation which (in the quasistatic limit) carries out any desired *idempotent* function g . (To emphasize the connection with results in stochastic thermodynamics, we go on to note that in the quasi-static limit, the entropy production incurred by our stochastic process vanishes.)
2. We now emphasize that the cited literature [Tatsuhiko Saito. Products of idempotents in finite full transformation semigroups. In *Semigroup forum*, volume 39, pages 295–309. Springer, 1989.] provides a *constructive* recipe for decomposing any non-invertible function into a composition of idempotent functions (in fact, a composition into the *minimal* number of idempotents). The same decomposition can also be applied to invertible functions, as long as one hidden state is available. This decomposition into idempotents can be combined with the construction in Appendix B to create CTMCs that achieve our space and time costs.
3. As before, we point out that if one has a sequence of idempotents $\{g_i\}$ that implements a desired function f , then by applying the construction in (1) to each of the idempotents g_i constructed in (2), one “joins end-to-end” a sequence of master equations to create an aggregate master equation that implements f .
4. We have introduced a detailed example, Example 2 (along with new Figure 2, illustrating it), which shows how to carry out the function $f : x \rightarrow x + 1 \pmod{16}$ over 16 states using 4 hidden states and the corresponding minimal number of timesteps (5). In the figure, we illustrate the decomposition of f into idempotents. This example more clearly demonstrates the constructive nature of our result.

As a final note, we agree with the reviewer that the fact that any logically-invertible function can be implemented with a master equation is not entirely surprising. However, we do not see this result as the main contribution of our work. Rather, we show that there is a fundamental --- and easily calculable --- tradeoff between the number of hidden states and hidden timesteps required to carry out any given function using a master equation. This, we argue, is in fact surprising and novel - as we now emphasize in our revised manuscript.

Moreover, the claim that the considered space/time tradeoff or relationship is of some physical relevance is actually not supported, though mentioned as one of the basic motivations of the paper.

We appreciate the reviewers suggestion. We have now updated the paragraph explaining the motivation behind our definitions of space and time costs, in terms of additional state space requirements in the physical system implementing the desired function f , the complexity of the control protocol needed by that system to implement f , and the time taken by the control protocol to do so:

“In the real world, often it will be costly to have many hidden states and / or many hidden timesteps. For example, in a real-world situation, increasing the number of hidden states generally requires adding additional storage capacity to the system, e.g., by using additional degrees of freedom. Similarly, increasing the number of hidden timesteps carries a “control cost”, i.e., it increases the complexity of the control protocol that is used to drive the dynamics of the system. Moreover, transitions from one timestep to the next, during which the set of allowed state-to-state transition changes, typically require either the raising or dropping of infinite energy barriers between states in some underlying phase space. Such operations typically require some minimal amount of time to be carried out. Accordingly, the minimal number of hidden states and the minimal number of hidden timesteps that are required to implement any given function f can be seen as novel “costs of computation” of a function f .”

While we believe such costs are relevant in many real-world situations, we have softened the text to not imply that such costs are *fundamental* from a physics point of view.

The authors suggest that their statements comply with previous work (Refs.[23-25]), e.g. when considering Example 1. But other new physical examples showing predictive power of the approach are missing.

Reference is always to quantum dots. The result remains too abstract.

We now develop a new concrete example, that is more elaborated than a quantum dot. Example 2 shows how to carry out the function $f(x) = x + 1 \text{ mod } 16$ over 16 states using 4 hidden states and the corresponding minimal number of timesteps (5). In the accompanying figure (Figure 2), we illustrate the decomposition of f into idempotents, providing an explicit construction for how to implement this f .

In the Discussion, we discuss further some physical aspects of our work, and in Appendix B, we discuss our constructions in the context of entropy production, as defined in stochastic thermodynamics (all our constructions can be realized with no entropy production). However, we emphasize that although our motivation was drawn from statistical physics, our results are more generally applicable to any context where master equations arise.

As a final comment, we wholeheartedly agree that there are a large number of concrete systems that should be analyzed using our results. However, in the interests of space, we defer such analyses to future work.

Additionally, how to fix the initial conditions on the extra degrees of freedom?

Our implementations with hidden states will carry out the desired mapping over the visible states *no matter what the initial distribution over visible states is*, as long as the system initially occupies *some* visible state at $t=0$ with probability 1. We have added a paragraph in Section II (after Definition 4) explaining this in more detail.

Is there any estimate on the extra dimensions needed?

Indeed; our main result states precisely how many extra states are needed to carry out a given f with a certain number of timesteps, and conversely how many timesteps are needed to carry out a given f with a certain number of hidden states. (These extra states need not be indexed in any particular number of dimensions.) To emphasize this finding, we highlight it in the last paragraph of the new Introduction:

“Our main results are exact expressions for the minimal number of hidden states needed to implement a single-valued function f , and the minimal number of hidden timesteps needed to implement f given a certain number of hidden states.”

The second point is about the treatment of time-inhomogeneous situations as mentioned.

Despite mentioning time-inhomogeneous master equations, there is no actual connection to this topic.

The basic result on which the paper stands is drawn from Ref.[42], which refers to semigroups (actually in a framework quite different from the standard one for time-homogeneous master equations considered for physical systems, so that in any case the point should be clearly spelled out to address a broad readership). The extension to the time-inhomogeneous case is certainly not a trivial step, but it is not mentioned. Indeed time is never mentioned, nor taken explicitly into account.

In our manuscript, we study the problem of implementing a function $f : X \rightarrow X$ using a master equation. Moreover, all of our results, such as the minimal number of hidden states and number of timesteps required to carry out an f , *require* the use of time-inhomogeneous master equations. In fact, most f cannot be implemented at all using time-homogeneous chains -- a result which was implicit in our theorems, but was not explicitly stated. To clarify and emphasize these points:

1. We now explicitly emphasize that we do not assume time-homogeneous CTMCs in section II A.

2. We now provide explicit constructions (in Appendix B), showing how our bounds can be achieved with actual CTMCs. These CTMCs are time-inhomogeneous, and indeed **must be so** in order to implement most functions (as we now emphasize).

Regarding the reviewer's comment about our use of semigroup theory: we use semigroup theory to analyze the semigroup associated with the function f (i.e., the function to implement), rather than the continuous-time master equation. Nonetheless, we uncover the surprising result that the properties of f (in particular, algebraic properties of the semigroup defined by f) place constraints on the kinds of master equation which can implement f .

We agree with the reviewer that our use of semigroup theory is quite uncommon in the context of analyzing master equations. However, we believe that this novel connection between two disparate mathematical fields is one of the important and novel contributions of our manuscript.

When considering the composition of time-dependent maps, also continuity becomes an important issue in order to comply with the dynamics of a physical system, so that this is not a trivial point. The authors should definitely clarify this point.

We appreciate the reviewer's suggestion to clarify the continuity (or lack thereof) of the various quantities we consider. We have now provided a construction (in Appendix B) that shows how any idempotent function can be carried out by an explicitly-specified CTMC. In this Appendix, we also show that this construction transforms the probability distribution over states in a *continuous* manner, starting from any desired initial distribution. As described in the Appendix, by "gluing together" (in time) multiple such constructions, and choosing the desired initial distributions appropriately, one can construct CTMCs that achieve our space and timestep bounds while always changing probabilities continuously.

Reviewer 4 comments and responses:

In this manuscript Wolpert et al. discuss the "space cost" and "time cost" in order to implement a single valued function with a master equation. While the general subject of the manuscript is interesting, I find the manuscript really hard to follow. The main problem in my view is the lack of clear definitions for many of the quantities introduced by the authors.

I would expect that even specialists would have a hard time reading this manuscript. I stopped after the end of Sec. II A.

Several definitions remain unclear to me.

There is a lot of jargon and almost nothing is defined in an equation.

It remains to be seen whether their results deserve publication in Nature Communications. Specific considerations follow.

We appreciate the reviewer's suggestions. We have significantly reorganized Section II of the manuscript so as to improve clarity. In particular,

1. We have added a section (Section IIA) where many simple definitions are now provided. Section IIB now contains the definitions of space cost and time cost (these used to be in Section IIA).
2. We have rewritten the text so as to eliminate some terms that we realized are not absolutely required. Remaining terms are defined in numbered definitions and, where appropriate, equations.

1) Overall the manuscript is really heavy on jargon. I would recommend that the authors eliminate some of it. For example, you can eliminate "time-inhomogeneous master equation" and simply say "master equation" in the introduction.

We have edited the manuscript to simplify and remove jargon where possible. In particular, as mentioned, we have eliminated all use of certain terms after confirming that we could avoid using them.

2) In the introduction, it is crucial that the authors clearly explain what is a logically invertible function and a logically non-invertible function. Do you need the word "logically"?

By these terms, we meant invertible / non-invertible in the usual mathematical sense for functions. To clarify, we have dropped the word "logically", which we agree was superfluous.

3) In the introduction, "Given this" and "Motivated by this" might not be the best way to start a paragraph.

Many thanks, we have taken this stylistic suggestion.

4) In the introduction, change "out discussion" to "our discussion".

Many thanks, corrected.

5) On page 2, in the paragraph starting with "Both these" the authors say that "changing energy barriers involves doing thermodynamic work". That is not correct, changing the energy of a state does involve work but changing the energy barriers between states does not. Also the statement "everything else being equal" is quite unclear.

We agree with the reviewer that changing energy barriers between states can be done with no work, and have corrected our manuscript. In fact, we have rewritten our discussion of the real-world implications of space and time costs, particularly in the following paragraph in the Introduction:

“In the real world, often it will be costly to have many hidden states and / or many hidden timesteps. For example, in a real-world situation, increasing the number of hidden states generally requires adding additional storage capacity to the system, e.g., by using additional degrees of freedom. Similarly, increasing the number of hidden timesteps carries a “control cost”, i.e., it increases the complexity of the control protocol that is used to drive the dynamics of the system. Moreover, transitions from one timestep to the next, during which the set of allowed state-to-state transition changes, typically require either the raising or dropping of infinite energy barriers between states in some underlying phase space. Such operations typically require some minimal amount of time to be carried out. Accordingly, the minimal number of hidden states and the minimal number of hidden timesteps that are required to implement any given function f can be seen as novel “costs of computation” of a function f .”

6) "minimal number of hidden states of hidden time-steps" looks wrong.

Many thanks, corrected.

7) The title of subsection IIIA is not good since the authors also define space cost in this part. Adding structure to the manuscript by dividing this subsection will probably improve presentation.

We have re-organized our manuscript, including updating the title of this subsection, as suggested by the reviewer.

8) All definitions in Sec IIIA would need an equation, otherwise it is really hard to follow: after a page, one is left with a lot of different names that cannot be identified with an equation with a number. Examples of things that have to be defined in equations with numbers are: adjacency matrix, embedable, limit-embedable, space cost, restriction of a matrix, stable matrix, and time-step cost.

We have followed the reviewer's suggestions and defined each term in its own numbered Definition, accompanied by an equation where appropriate.

Reviewers' comments:

Reviewer #2 (Remarks to the Author):

The authors have revised the paper in an important way, especially trying to pinpoint many aspects that in the first version were too loosely mentioned.

One of the main points, namely the actual connection with time-dependent master equations, has been now considered in detail in the Appendix. There is one point that I think still deserves clarification. It was actually already present in the first version, but now appears to be crucial. In Appendix A and B it appears that the rate matrix introduced to construct the master equation does depend on the initial condition. How can this be? The master equation should reasonably only be determined by the function it is meant to reproduce over a given time.

Before the paper can be considered for publication I find the authors should clarify this issue.

Response to Reviewers

There is one point that I think still deserves clarification. It was actually already present in the first version, but now appears to be crucial. In Appendix A and B it appears that the rate matrix introduced to construct the master equation does depend on the initial condition. How can this be? The master equation should reasonably only be determined by the function it is meant to reproduce over a given time.

We thank the reviewer for their careful reading and agree that this deserves some clarification.

For the purposes of simply implementing a conditional distribution P (the primary subject of the paper), the q we use to construct a sequence of rate matrices can be any distribution whatsoever. If we evolve any initial distribution over states $p(0)$ according to that sequence of matrices, the effect will be to apply the desired conditional distribution P to $p(0)$. This is true whether or not $p(0) = q$. (To see this, note that the conditional distribution implemented by any master equation depends only on the sequence of rate matrices $Q(t)$, but never on the initial distribution $p(0)$.) So as far as implementing P is concerned, q is just a parameter of our rate-matrix construction procedure.

We did make this point (briefly) in the paragraph of Appendix B beginning, "Thus, all the conditions given in Definition 6 concerning...". However, that parameter q of our rate-matrix construction procedures happens to specify the equilibrium distribution of that master equation at $t=0$. This means that if $p(0) = q$ --- that is, if we initially start the master equation in equilibrium --- then it turns out that in addition to implementing P , we get two other desirable properties: (1) $p(t)$ is a continuous function of t , and (2) our constructions incur zero entropy production. Unfortunately, in Appendix A especially, we conflated i) the role of q as an arbitrary parameter for the construction of a sequence of rate matrices to implement P ; and ii) the fact that running that sequence of rate matrices in the case that $p(0)=q$ happens to result in these two properties. Not surprisingly, this led to confusion. To clarify these points, we made the following modifications to the manuscript:

1. In the Discussion section, we have added a parenthesized sentence: "Nonetheless, in Appendix B, we show by construction that one can implement any f using the minimal number of hidden states and timesteps using a master equation that (1) obeys detailed balance, (2) evolves probabilities in a continuous manner, and (3) has vanishing entropy production, i.e., is thermodynamically reversible. The latter two properties are satisfied when the equilibrium distribution of the master equation at $t=0$ (determined by the choice of q in the construction in Appendix B) coincides with the initial distribution over states (this and related issues are studied further in [38])."

2. We have fixed Appendix A, to clarify the distinction between q and the initial distribution $p(0)$ and to fix a minor mistake.

3. We have emphasized the distinction between q and the initial distribution $p(0)$ in Appendix B. We also re-emphasized that when $q=p(0)$, our construction achieves zero entropy production, and provides that $p(t)$ is a continuous function of t .

REVIEWERS' COMMENTS:

Reviewer #2 (Remarks to the Author):

In my view the authors have revised the paper so as to clarify the point raised.

“Reviewer #2 (Remarks to the Author):

In my view the authors have revised the paper so as to clarify the point raised.”

There were no further comments to respond to. We thank the reviewer very much for their time.

Best regards,

David H Wolpert
Artemy Kolchinsky
Jeremy A Owen